# High density optical neuroimaging predicts surgeons's subjective experience and skill levels

**Hasan Onur Keles**[1]*, **Canberk Cengiz**[2], **Irem Demiral**[3], **Mehmet Mahir Ozmen**[4], **Ahmet Omurtag**[5]

1 Department of Biomedical Engineering, Ankara University, Ankara, Turkey, 2 Department of Electroneurophysiology, Istinye University, Istanbul, Turkey, 3 Department of OB&GYN, 29 May State Hospital, Ankara, Turkey, 4 Department of Surgery, Istinye University, Istanbul, Turkey, 5 Department of Engineering, Nottingham Trent University, Nottingham, United Kingdom

* hokeles@ankara.edu.tr

**Data Availability Statement:** All relevant data are within the manuscript and its Supporting information files.

## Abstract

Measuring cognitive load is important for surgical education and patient safety. Traditional approaches of measuring cognitive load of surgeons utilise behavioural metrics to measure performance and surveys and questionnaires to collect reports of subjective experience. These have disadvantages such as sporadic data, occasionally intrusive methodologies, subjective or misleading self-reporting. In addition, traditional approaches use subjective metrics that cannot distinguish between skill levels. Functional neuroimaging data was collected using a high density, wireless NIRS device from sixteen surgeons (11 attending surgeons and 5 surgery resident) and 17 students while they performed two laparoscopic tasks (Peg transfer and String pass). Participant's subjective mental load was assessed using the NASA-TLX survey. Machine learning approaches were used for predicting the subjective experience and skill levels. The Prefrontal cortex (PFC) activations were greater in students who reported higher-than-median task load, as measured by the NASA-TLX survey. However in the case of attending surgeons the opposite tendency was observed, namely higher activations in the lower v higher task loaded subjects. We found that response was greater in the left PFC of students particularly near the dorso- and ventrolateral areas. We quantified the ability of PFC activation to predict the differences in skill and task load using machine learning while focussing on the effects of NIRS channel separation distance on the results. Our results showed that the classification of skill level and subjective task load could be predicted based on PFC activation with an accuracy of nearly 90%. Our finding shows that there is sufficient information available in the optical signals to make accurate predictions about the surgeons' subjective experiences and skill levels. The high accuracy of results is encouraging and suggest the integration of the strategy developed in this study as a promising approach to design automated, more accurate and objective evaluation methods.

**Funding:** The author(s) received no specific funding for this work.

**Competing interests:** The authors have declared that no competing interests exist.

## Introduction

Excessive workload or acute stress may impact surgeons' ability to process all the information available during surgery in the operation room (OR) and may result in low situational and safety awareness, impaired decision-making and performance. Under excessive workload, a surgeon may be more easily distracted, entertain fewer alternatives, or persist with ineffective strategies [1–3]. The extent to which mental workload degrades performance depends upon surgeons' experience. Expertise is characterized by a combination of high performance and low cognitive load, allowing expert surgeons to process larger amounts of information and respond appropriately to unexpected events [4]. The mastery of complex tasks correlates with a progressive decrease in mental workload [5, 6] and minimally invasive operations are more workload intensive than open operations [7].

Several methods have been used to measure workload in surgery, including, subjective rating scales and physiological measurements (EEG, EKG etc) [8]. One of the most widely used is the subjective ratings scales known as NASA-TLX [9–11]. It is a multidimensional scale, initially developed for the use in the aviation industry. The NASA-TLX provides an overall index of mental workload as well as the relative contributions of six subscales: mental, physical, and temporal task demands; and effort, frustration, and perceived performance. The NASA-TLX score on an interval scale ranging low (1) to high (20) for each subscale. The widespread use of NASA-TLX is associated with its simplicity of application and interpretation. However, the NASA TLX has been criticized for not measuring the mental workload in a real time [12, 13] NASA-TLX is filled out after task completion to gather participants' recall of their cognitive effort during surgery. Therefore, NASA TLX can be intrusive to primary task performance Thus, there is a need for more automated, more accurate and objective evaluation methods.

Psychophysiological measures allow a more objective workload assessment and can provide uninterrupted evaluation. They are gaining in popularity as progress in wearable sensor technology makes this approach less intrusive and capable of delivering continuous, multi-modal information. Electroencephalogram (EEG) and Heart Rate (HR), Heart Rate Variability(HRV) have also been correlated with NASA-TLX scores [14, 15] as well as expertise, task complexity and poor performance in surgery. Similarly, optical imaging (NIRS) has been used to assess the cognitive load of surgeons. It has been implemented to capture activation patterns in specific brain areas during surgical tasks with resulting correlations to surgical expertise and technical performance [16–18].

NIRS is a developing technology for assessing the hemodynamic activity of the human cortex. NIRS can be a portable and wireless. As a wearable and lightweight method, NIRS provides a safe and practical approach for monitoring surgeons' brain activity, and may be adopted even for applications in OR. Technical progress has also made it possible to use NIRS with a high density and multi-distance source detector separations [19]. NIRS is a method which is very sensitive to the superficial layers of the head, i.e. the skin and the skull, where systemic interference occurs. Thus, the NIRS signal is contaminated with systemic interference of superficial origin. Our approach to overcome this problem has been the use of additional short source-detector separation optodes as regressors. In this study, high density NIRS device allow us to investigate the hemodynamic changes in a different depth using different source-detector separations. We are able to use 52 channels with a separation distance of 1.5 cm, 36 with 2.12 cm, 68 with 3 cm, and 48 with 3.35 cm.

In this paper, we investigate cerebral hemodynamic correlates of NASA-TLX by segregating the ratings into sets of high and low scores. We also investigate the ability of prefrontal activations to discriminate between the subjects' subjective experiences (high v low NASA-TLX score) as well as their skill levels (Student v Attending), by using machine learning techniques.

Our aim is to show that there is sufficient information available in the optical signals to make accurate predictions about the surgeons' subjective experiences and skill levels. We focussed on how both the topographic location and the sampling depth of a channel (which depends on source detector separation) affected the information available regarding the cognitive load and the expertise of the subjects. We also examined the effects of superficial signal regression by using the signal from the shortest (1.5 cm) separated channels to minimize the non-cerebral component from that of the normal (3 cm) and long (3.35 cm) separated channels [20, 21]. Our automated approach can be used instead of currently established metrics used for certification in surgery. These results demonstrate that the combination of advanced fNIRS imaging with machine learning approaches offers a practical and quantitative method to predict subjective experience and skill levels. The reported optical neuroimaging methodology is well suited to provide quantitative and standardized metrics for professional certifications and surgery education.

## Methods

### Subjects

Sixteen surgeons (11 Attending surgeons and 5 surgery resident) and 17 medical students participated in this study. Surgery residents (5 surgery residents) are excluded from further analysis due to low sample number. Subject demographics are listed in S1 Table. To avoid any issues regarding hemisphere-specific activation, only right-handed participants were selected. All participants provided written informed consent prior to the study commencing. Participants had normal or corrected to-normal vision. The Ethical Committee of the College of Medicine at Medipol University (10840098–604.01.01-E.33230) approved the study.

### Experimental design

The study was conducted in a laboratory equipped with a Laparoscopic trainer box. Each trial took about 40 minutes including total time spent by participants to perform the tasks and setting up the system and devices. At the beginning of the trial, two 2-minutes-long videos that demonstrates the tasks were shown on a computer screen to the subjects. The rules and the possible errors were explained and demonstrated. The subjects were informed that they were free to stop and leave the experiment at any time they wished without risk of facing any circumstances. The surgical equipment and their usage were introduced to 20 of 33 subjects who had no laparoscopic surgery training or experience and were given 10-minutes free time to train on tasks and understand how the devices and equipment work. 13 of 33 subjects who had previous experience or training of laparoscopic surgery skipped this step.

The tasks of the experiment included two Fundamentals of Laparoscopic Surgery tasks, Peg Transfer and Threading as "Task 1" and "Task 2" respectively. Every task was preceded by 1-minute-long resting state fNIRS recording and followed by fulfilling of NASA-TLX questionnaire by the subjects to evaluate his/her own performance and taskload in that task. In addition to this, following the training session, NASA-TLX Pre-Test was fulfilled by subjects with no previous laparoscopic surgery experience or training.

Peg transfer task involved grasping, lifting and relocating rings from one rod to another using both laparoscopy graspers and was performed on a ring stack base. Four rods were selected and labelled 1, 2, 3 and 4, at the left-hand bottom, top left-hand, top right-hand and bottom right-hand corners on the ring stack base respectively. Four rings were initially put over rod 1 at the beginning of the trial. Participants were instructed to move each ring individually from rod 1 to 4. The procedure included picking up rings from rod 1 and place it to rod 2 one by one with left-hand only. Once all rings were moved to rod 2, they were moved

individually by grasping and lifting the ring up with the left-hand, passing the ring over to the right-hand, then placed onto rod 3 using the right-hand only. The procedure completed by moving the rings individually from rod 3 to 4 using right-hand only. One of the two defined error types in this task were dropping a ring during transfer steps except dropping it on the beginning or aimed rod. The other one is re-grasping a ring with right-hand grasper after dropping it while passing it over from left-hand to right-hand.

Threading involved grasping a piece of string and passing it through the holes using both graspers and was performed on a Threading Base. The start of a piece of string was initially placed on the left-hand side of the base and the participants had to start passing it through the holes which were labelled 1–7 in a zigzag pattern using both hands as they were willing to. Passing the string beside or behind a hole instead of through the hole and trying to continue on passing the string through the following holes due to the visual distortion (often caused by the 2-dimensional viewing on the screen) was a defined error type in this task.

Our experimental procedure is summarized in Fig 1. Following the introduction and training session, the subjects were given time to relax and prepare for the tasks. After this step, the NIRS device was placed on the subjects' heads. In order to have the best result, the inferior border of the device was placed over upper border of nasion and eyebrows, the hair that might be an obstacle for optodes to work efficiently were tried to be gently retracted and kept away and the string of the device was adjusted and fitted according to the verbal feedback from the subjects. This was followed by the calibration of the optodes for recording.

After the optimal calibration was determined, any sound, extra light source and any other attention distracting stimuli were removed from the experiment environment. At this point, the subjects who had a 10-minute-long training session were asked to fill NASA-TLX Pre-Test. Then, all the subjects were reminded not to talk during the NIRS recordings. Later, the subjects were asked to stand still with closed eyes for resting state recording until they were asked to start the task. NIRS recording started at this point. 60 seconds later, the resting state recording was stopped, the task recording was started and the subjects started Task 1. If the subject had completed the task in less than 6 minutes, the task was finished, NIRS recording was paused and the time was recorded. Otherwise the task was ended unfinished and recorded so, and again NIRS recording was paused. Following this, NASA-TLX form was given to the subjects.

After completing the NASA-TLX, the subject was asked to complete Task 2 with the same procedure of resting state and task NIRS recordings, time recordings and fulfilling of NASA-TLX forms respectively and the experiment was finished. The completion time of task is recorded for each subjects during the experiment.

## Optical imaging

Functional neuroimaging data was collected using a high density NIRS device (NIRSIT, OBE-LAB, Korea). This system has 24 laser source (780/850nm) and 32 photo detectors at a sampling rate of 8.138Hz. This device uses 204 channels in total attached to the forehead of the subjects for measurements. Each type of channel covers the areas coinciding with lower parts of the dorsolateral prefrontal, upper part of the orbitofrontal and medial prefrontal, and part of the ventrolateral prefrontal cortex.

The NIRS system that was picked for the experiment was capable of measuring signals from four source-detector (SD) separations: 15, 21.2, 30, and 33.5 mm, and this allowed the measurement of alterations at various depths. Movements were tracked in real-time by using a gyroscope and an accelerometer. The measurements were obtained from the prefrontal cortex, where the center of the lowermost optical probes was aligned to the frontal pole zero (FPz) location of the 10–20 EEG system to remove positional uncertainty between subjects.

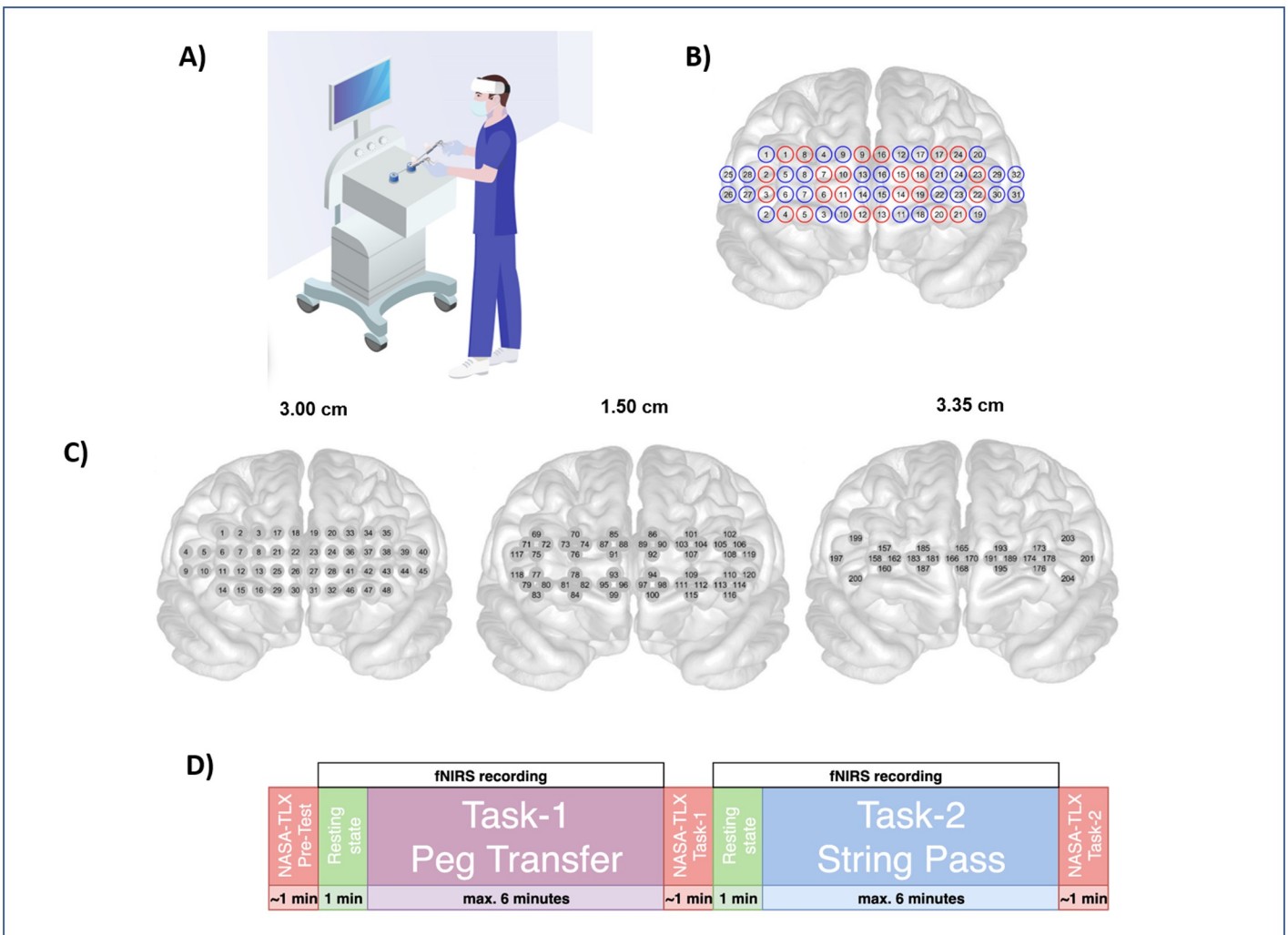

**Fig 1. Experiment description.** (A) Schematic depicting the laparoscopic box simulator where surgeons and students perform the FLS tasks. High density, wireless NIRS is used to measure functional brain activation. (B) Optode positions cover the frontal cortex (24 sources and 32 detectors). Red circles are sources, blue circles are detectors C) Channel numbers with different source-detector separations. Total 204 channels: 52 channels with a separation distance of 1.5 cm, 36 with 2.12 cm, 68 with 3 cm, and 48 with 3.35 cm D) Experimental protocol design. Schematic showing the experimental design for this study. All attending, resident and students performed the FLS tasks. After each Nirs recording, participant filled out the NASA-TLX.

## Data analysis

**Preprocessing.** The detector readings at two wavelengths for each channel were converted into concentration changes of oxy- and deoxy-hemoglobin by using the modified Beer-Lambert Law [22]. The differential path length factor (DPF) values for light wavelengths 780 nm and 850nm are 5.075 and 4.64 in respectively [23, 24].

The haemoglobin time series thus obtained contain information about the local brain activity as well as extraneous components, considered as artifacts, arising from non-cerebral tissue and muscle activations, respiration and heartbeat, as well as from systemic physiological effects such as Mayer waves. Other common artefacts are signal transients due to head movement. These may create temporary variations in optical coupling leading to excursions in the detector readings. A comprehensive review of techniques designed to detect and minimise such artifacts is provided in [25].

The changes in hemoglobin concentrations were band-pass filtered in the range 0.01–0.5 Hz in order to diminish components with characteristic durations briefer than ~2 s and greater than ~100 s. This eliminated the effects of respiration (~0.3 Hz), the heart beat (~1 Hz), helped reduce some motion artefacts that had sharp transients, and reduced the slow baseline drift [26]. The characteristic scale of Mayer waves, with a period of ~10 s, partly over-lap with task evoked hemodynamic responses hence we did not attempt to filter them out; however they were not expected to influence our results, as Mayer waves likely do not correlate with cognitive load [27].

Next we used windowed standard deviation to quantify the presence of motion artefact [28]. This approach focussed on motion artefacts associated with excursions greater than that from the concurrent physiological effects. Note that if the standard deviation of the motion artefact was smaller, this implied that it was not a severe artefact. We calculated the standard deviation in nonoverlapping 10 s windows, and the median absolute deviation (MAD) of the set of standard deviations for each channel. Any window whose standard deviation was greater than 4.5 MAD values away from the median was considered as an outlier and excluded from subsequent analysis.

To confirm the plausibility of this approach we visually inspected the signals from randomly selected segments of the recordings from multiple subjects and experimental conditions. The outlier tended to occur approximately around the same time as increases in the accelerations as measured by the headset. Virtually all severe deflections were captured while many outliers contained only mild fluctuations, suggesting that our criterion was conservative.

**Feature extraction and selection.** As an indicator of the local activations in the prefrontal cortex we computed the standard deviation of the oxyhemoglobin changes in each channel over 10 s windows. Because greater evoked hemodynamic response tends to increase the standard deviation of the signal in a window, we took these values as the measure of the prefrontal activation. Note that the window mean of the signal may not be as good an indicator of activation in some cases when the evoked response is brief and followed by a dip. By separately exploring other variables such as the window mean, skewness and kurtosis, we determined that the standard deviation was the best indicator for the types of analysis reported in this paper. The standard deviation or variance have frequently been used in machine learning studies with fNIRS [29–31]. Other feature extraction techniques were described in [32]. The activations were used for analysing the information available from the channels with different separation distances. For some calculations we used the signals from the channels with a separation of 1.5 cm to perform superficial signal regression (SSR) in the 3 and 3.35 cm channels, in accordance with the methodology in [33]. This was done in order to explore the ability of short separation channel signals to eliminate the signal component originating from layers above the cortex. The feature types were prioritised by using the Pearson correlation between the observations and the labels, a standard feature-selection technique [23 and references therein]. The prioritization was used to select a small subset from the full set of features in the classification. Other feature selection techniques were also explored but not adopted since they did not appear to improve the classification accuracy. These were the Minimum Redundance and Maximum Relevance and the Chi-square tests, as implemented by Matlab's functions*fscmrmr* and *fscchi2*, respectively (Matlab v.8.2.0.701; The MathWorks, Inc., Natick, Massachusetts, United States).

**Classification.** We examined the cerebral hemodynamic correlates of NASA-TLX by segregating the ratings into sets of high and low scores. For each subject and episode the mean score (average over the 6 dimensions of the NASA-TLX) was used. The high/low scores were discriminated by using the median score as a cut-off value. Separate cut-off values were used for the Student and Attending groups of subjects. The statistical significance of the difference

between two sets of values (e.g. high v low scores or Student v Attending subjects) was determined by means of the non-parametric Kolmogorov-Smirnov test.

We also examined the ability of prefrontal activations to discriminate between the subjects' subjective experiences (high v low NASA-TLX score) as well as their skill levels (Student v Attending), by using machine learning techniques. The scores or the skill levels were used as the binary valued labels to be classified. For this purpose feature matrices were built with each row (an observation) representing the standard deviations averaged over an episode (e.g. Task 1) for a subject. Each column of the feature matrix corresponded to a channel. The columns therefore represented types of features available. We then chose a small group of features from the prioritised list of features and used it to train Support Vector Machines (SVM) with linear kernels. The accuracy of the SVMs were determined by means of 5-fold cross validation. Note that because each row is an entire experimental episode for a subject, the data from a subject that is in the training set was automatically excluded from the test set. This process was repeated 2000 times to obtain a distribution of accuracy values, revealing the variability due to different training-test partitions. The small set of features was then enlarged by progressively including more types of features and repeating the classification. This procedure allowed us to examine the accuracy as a function of the number of features used, and to focus on particular subsets of the local activations.

## Statistical analysis

In order to assess the statistical significance of the difference between two groups of paired results, we used the non-parametric Wilcoxon signed-rank test. The descriptive results (Figs 2–4) comparing two groups, such as low v high cognitive load or student v attending subjects, contained paired data for each channel. The null hypothesis was that the results from both groups were drawn from the same population. In testing this hypothesis we utilized the Bonferroni procedure which insured that if each of the $k$ tests has $p < a \, / \, k$, then the null hypothesis will be falsely rejected with a probability no greater than $_a$. We set $a = 0.05$; and $k = 16$, the product of the four types of channel separations and the two prefrontal sides (right v left). In assessing the statistical significance of the differences in the topographic representation (Fig 5) we consulted the corresponding box plots in Fig 4 as explained in the Results. The statistical significance of the accuracy of prediction was determined through the permutation technique [34]. In this technique we randomly shuffled the labels and reassigned them to the observations, thereby creating a surrogate set of data. This was then used in classification following the steps of feature selection and cross validation, and generated a null distribution of accuracy values. In order avoid crowding the plots with indications statistical significance, in presenting the results of classification (Figs 6–8) we show the null and actual distributions as shaded areas so that the ranges where they were sufficiently distinct (and likely statistically significantly different) were visually evident.

## Results

In this section we present the results of our analysis of the fNIRS data recorded from Student and Attending subjects. We hypothesized that the task related activations of PFC (measured by using changes in the optical signal as described in Methods) would reflect the subjective experience of the subjects as well as their levels of skill and, furthermore, that there would be differences in the activations due to the different sampling depths of the channels with different separation distances. The subjective experiences were monitored using responses to the NASA-TLX questionnaire. We considered subjects in two groups of skill levels: Students who had no laparoscopy or training experience and Attending residents who had previously

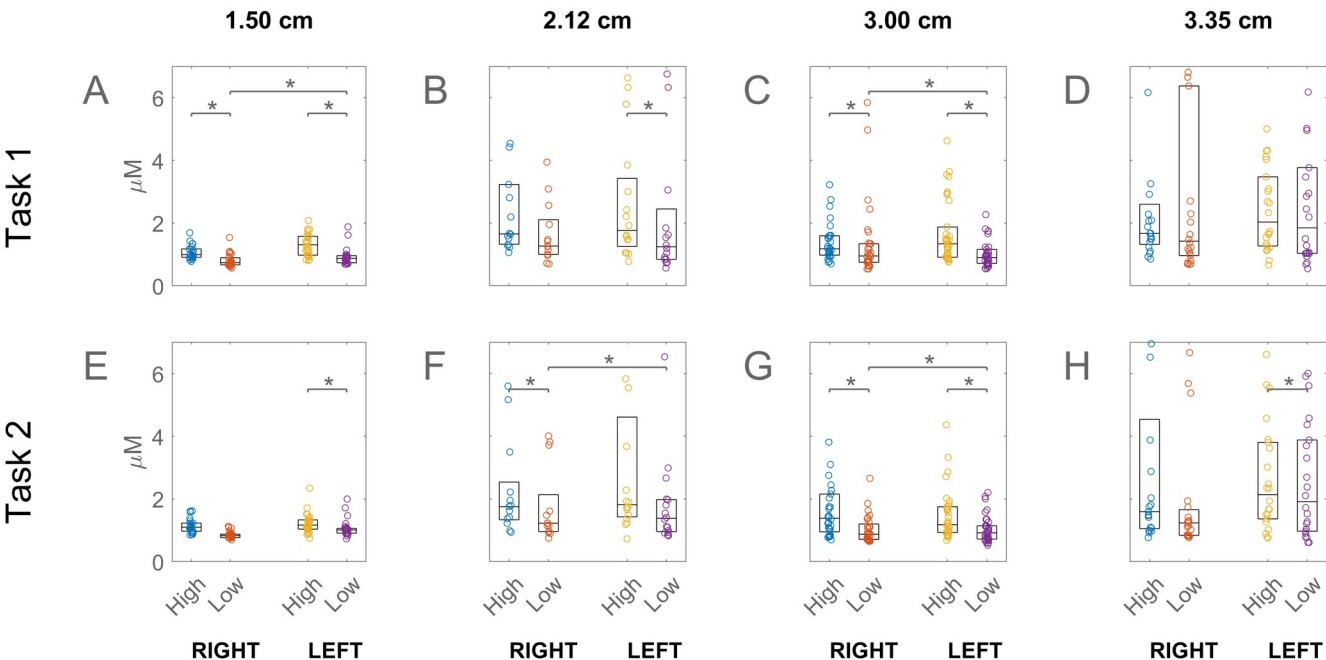

**Fig 2. Prefrontal activation associated with high and low NASA-TLX scores.** Channel separation distance for each column are labelled at the top of the figure. Student subjects only are shown. The high/low cut-off was the median score. On each box, the central mark indicates the median while the bottom and top edges indicate the 25th and 75th percentiles. Circles represent individual channels. Statistically significant difference between high and low scores is indicated by an asterisk above the boxes. The right and left prefrontal activations are shown separately as labelled at the bottom of the plot. The results for task 1 and 2 are shown in the top and bottom rows. For the values shown in this figure, please see S2 Table.

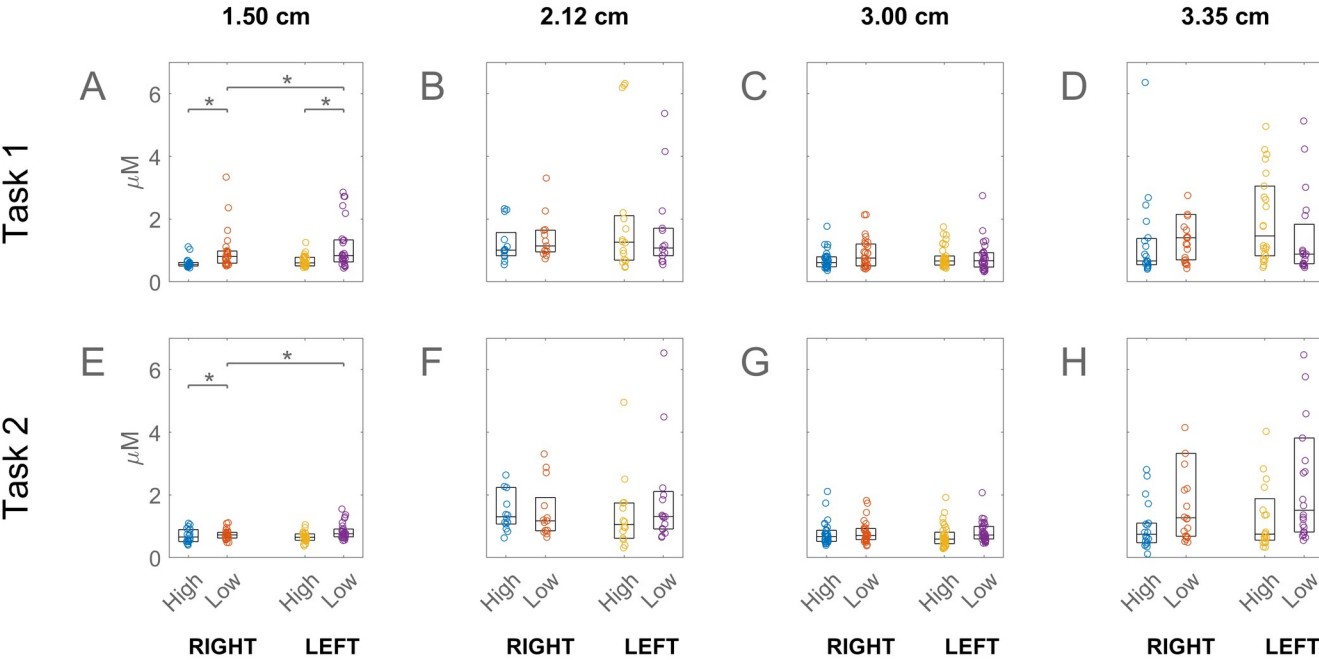

**Fig 3. Prefrontal activations associated with high and low NASA-TLX scores.** This figure contains the same results as in Fig 1, but it is for the attending subjects. For the values shown in this figure, please see S3 Table.

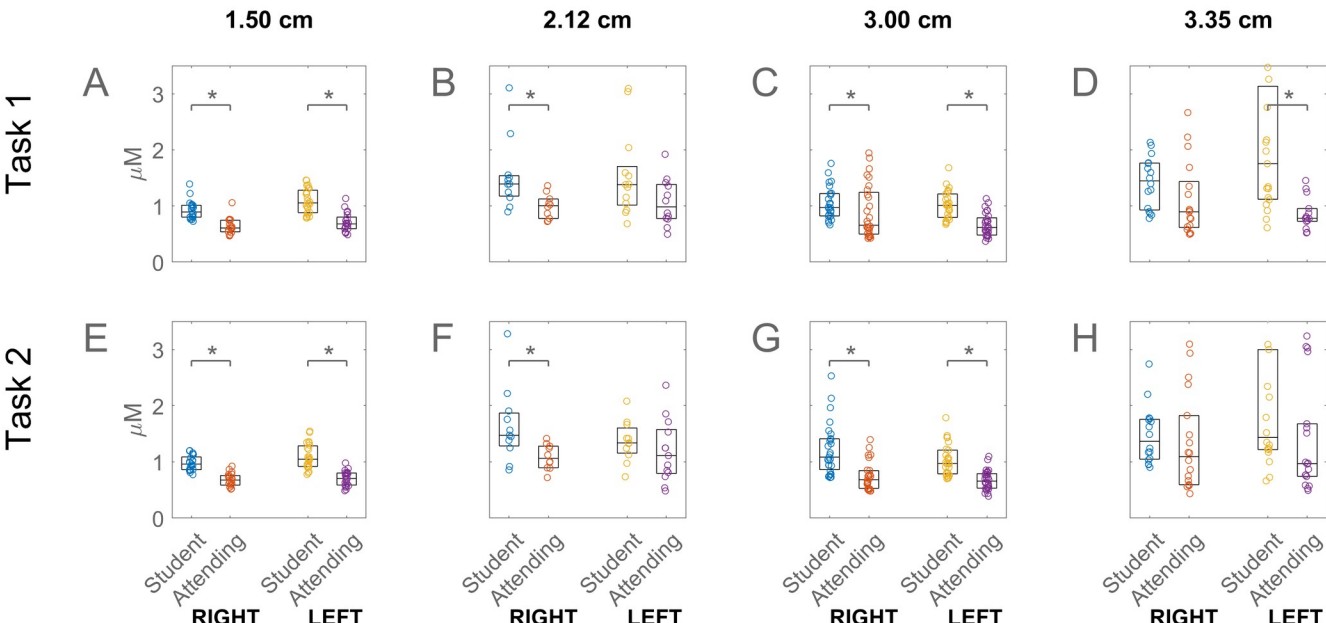

**Fig 4. Prefrontal activations associated with student and attending subjects for task 1 and task 2.** For the values shown in this figure, please see S4 Table.

performed a median of 75 laparoscopic operations (Table 1). We first compared the high v low NASA-TLX scoring subjects' activations (Figs 2 & 3), then the effects of skill level (Fig 4), and the topographic distributions of the differences due to skill level (Fig 5). We next used automated classification in order to quantify the ability of the fNIRS signals to predict the subjective experience during the training tasks as well as the skill level of the subjects (Figs 6 and 7).

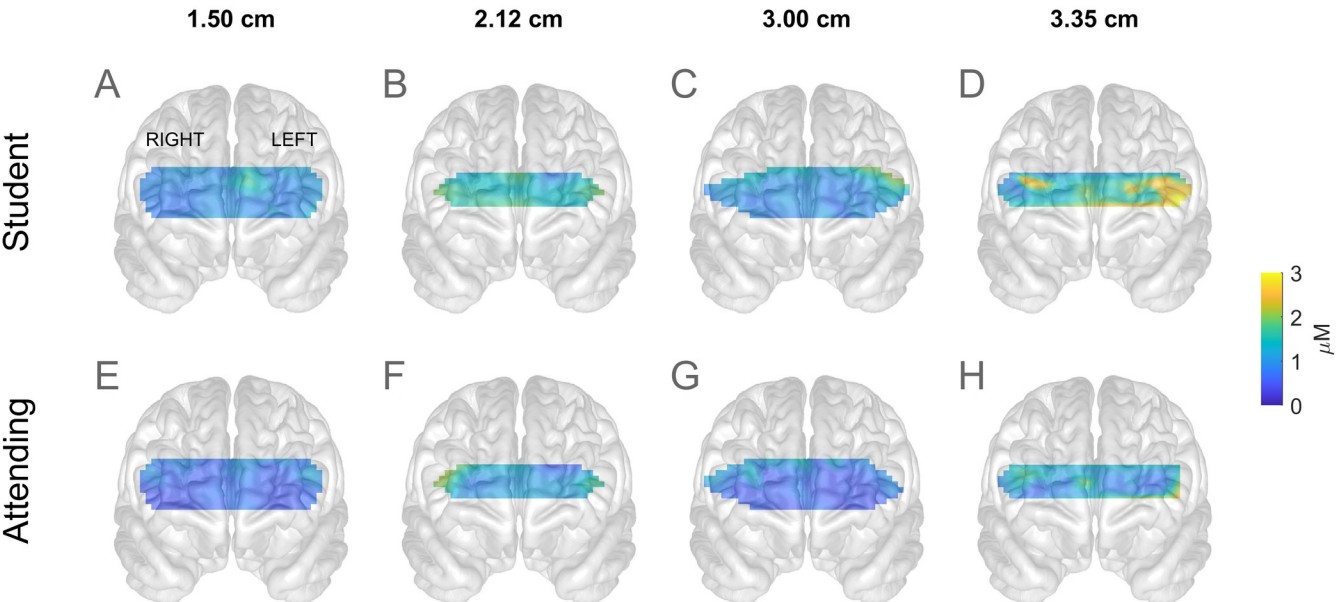

**Fig 5. Topographic projection of the prefrontal activations in task 1, for the student (top row) and attending (bottom) subjects.** Different channel separation distances are shown as different columns as labelled at the top of the figure.

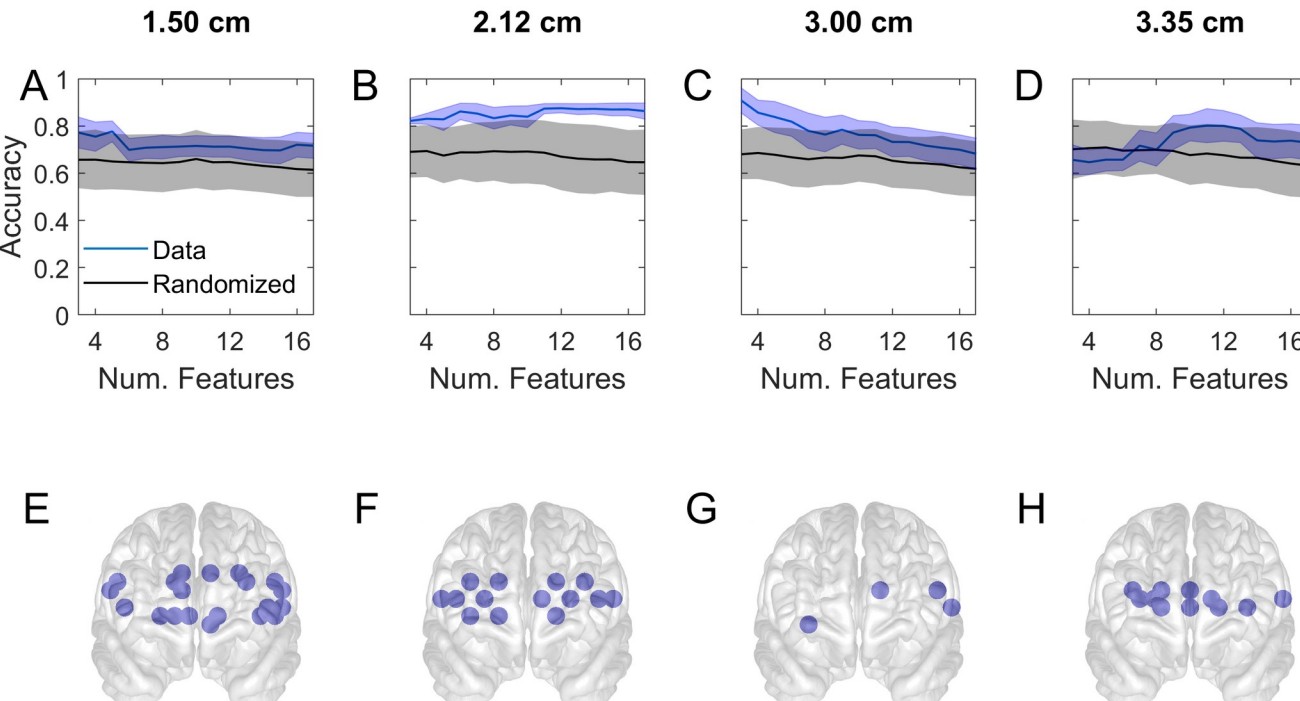

**Fig 6. Accuracy of classification of the NASA-TLX score of student subjects in task 1.** The scores were segregated into sets of high and low values by using the median score as cut-off and predicted based on prefrontal activations by using support vector machine. A-D show the accuracy as a function of the number of channels used. The thick curves are the mean of 2000 repetitions of different 5-fold cross-validations. The shaded region indicates the standard deviation of the variability in the accuracy. The results (blue curve) are compared with those from a surrogate set that contained the same data but the scores were randomly reassigned prior to classification (black). E-H show the frontal view locations of selected channels that resulted in high accuracy.

## Group differences

Fig 2 shows that in the Student subjects who experienced higher task load also had higher PFC activations. The figure shows each channel as a circle and the box indicates the range from the 25th to the 75th percentiles, with the line inside box showing the median. The y-axis is the standard deviation of the oxyhemoglobin changes averaged over an episode, such as Task 1 or 2, and over subjects. The variability of the response appeared lowest in the shortest and greatest in the longest separation (3.35 cm, D and H) channels. The figure shows that the right and left PFC differences in 3 cm channels was significant in the low load students.

Fig 3 presents the corresponding results this time for the Attending subjects. The figure indicates that while there was a similar tendency for the variability in the response to increase with channel separation, the difference between the high and low task loaded subjects was reversed relative to those of Student subjects. The Attending subjects who reported higher task load generally had lower PFC activations, the difference being statistically significant only in the shortest separations in Task 1 in the Right PFC.

Having examined the fNIRS correlates of task load, we turned to the effects of the differences in skill level. Fig 4 shows that lack of prior laparoscopy experience correlated with higher PFC activation. The students has significantly higher activations than Attending residents for most of the different sampling depths. In addition, for the student subjects there was a pronounced asymmetry in the case of the deepest sampling channel (D and H), the activation on the left being higher than on the right, although the asymmetry did not reach statistical significance.

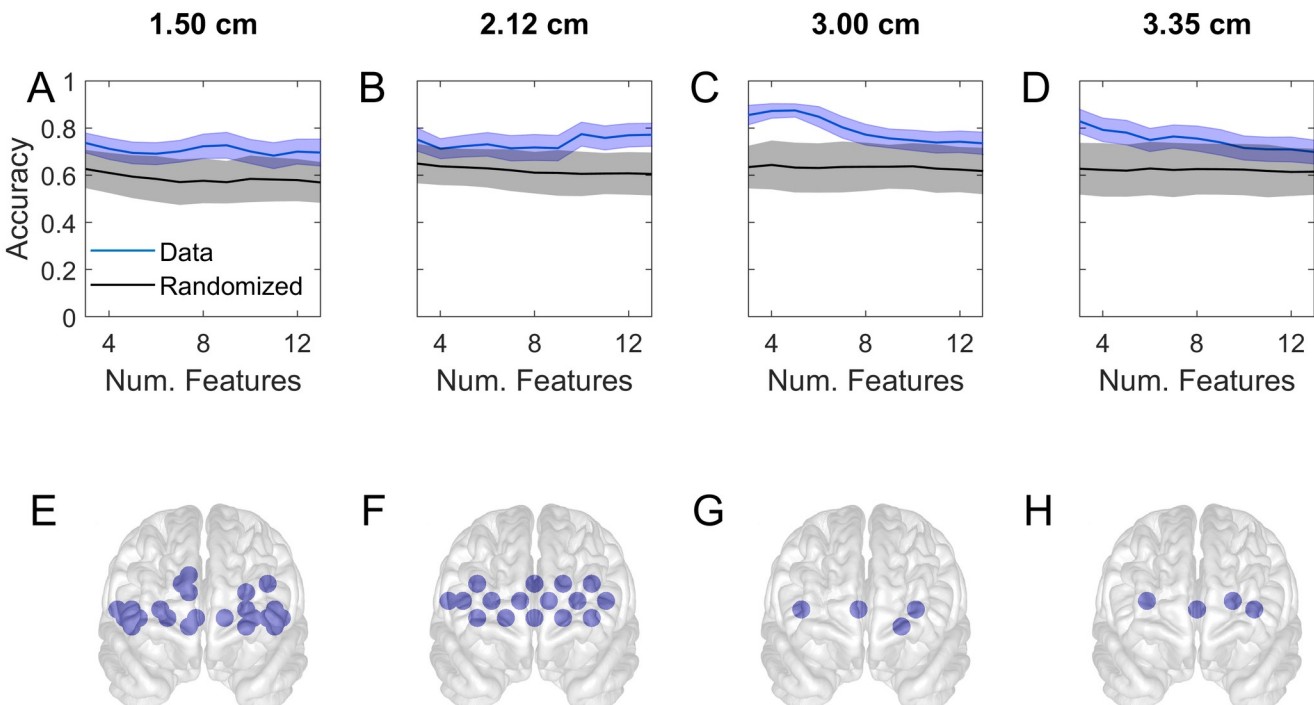

**Fig 7. Accuracy of classifying subjects into student v attending in task 1.** The subplots represent the same kind of information as in Fig 5.

The box plots so far presented show the values obtained from each channel, however it is instructive to see the locations of the individual channels together with their activations. To that end we present in Fig 4, the colour coded frontal view of activation projected on a drawing of the brain in order to illustrate the approximate locations. They were interpolated between the channels to show the spatial changes over a continuous field of activation. This figure only

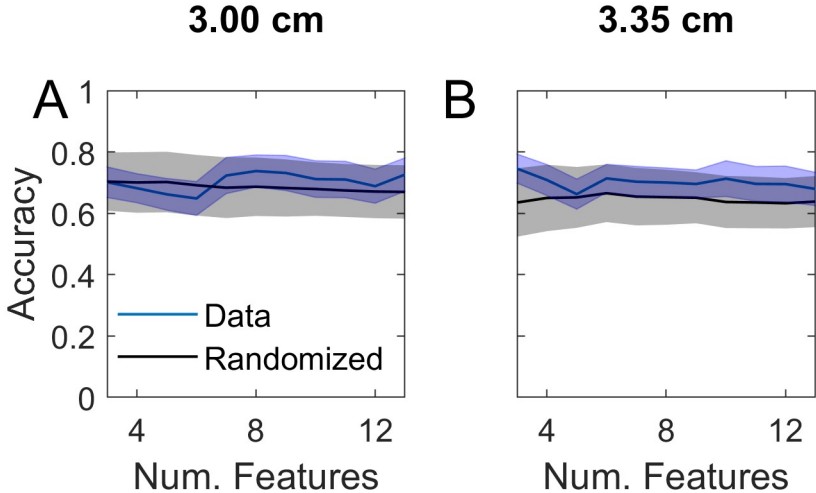

**Fig 8. Accuracy of classifying subjects into student v attending in task 1 for channel separations of 3 and 3.35 cm, using superficial signal regression (SSR) from the 1.5 cm channels.** This figure is the counterpart to subplots C-D of Fig 6 using SSR. The channel locations in C are not shown since the accuracy was low overall.

**Table 1. Subject demographics and descriptive data.**

| Group Demographics | | | | | | | |
| --- | --- | --- | --- | --- | --- | --- | --- |
| Group | Number | Median SE In Years (Range) | SE In Years Mean ±SD | Median Age (Range) | Age Mean ±SD | Median LSE In Number (Range) | LSE In Number Mean ±SD |
| Undergraduate Student | 17 | 0(0–0) | 0±0 | 19 (18–27) | 19.8±2.2 | 0(0–0) | 0±0 |
| Surgery Resident | 5 | 1.5(0,2–3) | 1.8±1 | 28 (26–29) | 27.6±1 | 0(0–40) | 9±15.6 |
| Attending Surgeon | 11 | 12(5–30) | 12±6.8 | 37 (28–55) | 37.4±7 | 75(8–350) | 133±120.8) |

SE:Surgery Experience; LSE: Laparoscopic surgery experience.

presents the results for Task 1. In the short channels (A, E, B, F) the difference between student and Attending subjects appears as a small difference in the tone distributed over the entire region. E.g. the blue in A and B are somewhat lighter than in E and F. That difference is the counterpart of the relatively shifted locations of the boxes in Fig 4A & 4B.

In the normal channel (C and G) there is a hint of high activation localized near the top left, within the dorsolateral PFC. The localization of the higher activation in student subjects is most apparent in the deep sampling channel (D and H). In student subjects (D), higher activation is visible by inspection in parts of the left lateral PFC and the orbitofrontal cortex. Some activation is localized near the lower part of the right dorsolateral PFC, although lower than that in the left.

To evaluate the statistical significance of the differences between the student (first row) and attending (2nd row) subjects in Fig 5, we used the corresponding box plots in Fig 4. Consider, for example, the higher PFC activation in the students relative to the attending population, namely the yellow regions in Fig 5D contrasted with the blue colour of the same regions in Fig 5H. This difference was significant because, as shown by box plots in Fig 4D, the activation in the student population in the left PFC was significantly different than that in the attending population.

## Machine learning results

The results so far showed the differences in PFC activations of subjects who reported different cognitive load and had different skill levels. In order to delve deeper into these differences, and quantify the amount of information that PFC activation may contain with regard to subjective experience, we used machine learning techniques. Using progressively greater numbers of features (prioritized as described in Methods) to predict subjects' NASA-TLX score led to the accuracies shown in Fig 6. As before, each column shows the results for a different channel separation distance. The top row of the figure (A-D) indicates that the accuracy and its range of variability under repeated cross validation (blue curve and shaded) are greater than expected by chance alone (black curve and gray shaded). In the short separation channels (A-B) the accuracy depends relatively little on the number of features included in classification. In B, the accuracy rises slightly with increasing number of features.

By contrast in the normal separation channel (Fig 6C) the accuracy is high in a small system and decreases quickly with increasing number of features. In order to visualize the locations of the channels which are responsible for the highest accuracy, we show their locations in the topographic plots directly underneath. For example G shows the four 3 cm separated channels that are in the small system whose accuracy is shows to be nearly 90% in C. The accuracy of the deep sampling channel (D) peaks at around 12 channels, whose locations are shown in H.

For the short separation channels the accuracy plots (A-B) are relatively flat, therefore we show a larger number of channels (the first 20) from the prioritised list (E-F).

Fig 7 gives the corresponding results for the prediction of skill levels. The figure indicates that the classifying ability of the short channels are spatially distributed (as in the previous figure), producing a relatively flat curve for the accuracy as a function of the number of features (A-B). This is reflected in the wide coverage of the channel locations of the first 20 channels (E-F). The normal separation channel gives the highest classification accuracy for small systems (C) and the locations of the first 4 channels are shown (G). As distinct from Fig 6 which was for classifying task load, Fig 7 shows that in the case of classifying skill levels only a small number of deep sampling channels (H) participate in generating the highest accuracy (D).

Some of the results we presented above were from Task 1 only (Figs 4–7). The counterparts of these results from Task 2 were on the whole similar with only minor differences, and were not shown.

## Discussion

In this paper we used high-density continuous-wave fNIRS data recorded from surgery students and Attending residents to show the extent of group differences in PFC activation between subjects with different levels of skill and subjective task load. We quantified the ability of PFC activation to predict the differences in skill and task load while focussing on the effects of fNIRS channel separation distance on the results. Our recordings were carried out using 52 channels with a separation distance of 1.5 cm, 36 with 2.12 cm, 68 with 3 cm, and 48 with 3.35 cm, each type of channel covering the areas coinciding with lower parts of the dorsolateral prefrontal, upper part of the orbitofrontal and medial prefrontal, and part of the ventrolateral prefrontal cortex (Fig 1). Our measure of PFC activation was based on the optically detected variability in the local changes in oxyhemoglobin concentrations. We also examined the effects of superficial signal regression by using the signal from the shortest (1.5 cm) separated channels to minimize the non-cerebral component from that of the normal (3 cm) and long (3.35 cm) separated channels.

The PFC activation was greater in students who reported higher-than-median task load, as measured by the NASA-TLX survey. This difference, visible in most channel separations, was statistically significant in all the 3 cm separated channels (Fig 2). Higher engagement of the PFC with greater task load is well known from previous studies [e.g. 25]. However in the case of Attending subjects the opposite tendency was observed, namely higher activations in the lower v higher task loaded subjects. This reversal was statistically significant in some of the 1.5 cm separated channels (Fig 3). This change in skilled subjects' PFC response relative to that of unskilled ones may have been due to the fact that parts of our experimental procedure and the stylized tasks unavoidably differed from the actual laparoscopic operations to which the skilled participants were accustomed. In such non-optimal situations the experts' established schemas are sometimes not applicable leading to higher cognitive load than in novices, in what has been called the expertise reversal effect [35].

We found that response was greater in the left PFC of students (Fig 4D and 4H), particularly near the dorso- and ventrolateral areas (Fig 5D), which was in accord with the known dominance of the left hemisphere in motor action. Previous studies have linked left hemisphere to behavioural efficiency, regardless of the subjects' handedness [36], as well as to interference processing [37], the neural representation of grasping [38], bimanual coordination [39], and overall movement organisation and selection [40]. Interestingly, localisation and lateralization arose in our data only in the deeper sampling channels (with a separation distance of 3.35 cm) clearly suggesting their cerebral origin. This asymmetry did not arise in Attending

subjects presumably because the skilled activity in their case had been relatively automated and its control shifted to non-frontal cortical areas [41, 42]. Note that the greater access to cerebral activity came at the cost of lower signal-to-noise ratio, reflected in the increasing response variability with channel separation distance (Fig 4A–4D). Several studies have used source detector separation distances in the range 2.5–3.5 cm for interrogating brain activity, and our longest separation is within this range [43–45].

Our results showed that the classification of skill level and subjective task load could be predicted based on PFC activation. In a machine learning study, the activations were first prioritised in terms of their Pearson correlation with the prediction targets (student v Attending or high v low subjective score), a small subset of the highest priority features (or channels) were selected for cross-validated prediction of the targets using SVM, then the cross-validated prediction was repeated by progressively increasing the number of features. The resulting accuracy of prediction (y-axis) as a function of the number of channels (x-axis) are shown for subjective task load in Fig 6 and for skill level in Fig 7. Repeated calculations with different training/test partitioning of the data was a cause of variability in the results. We used this variability to generate a distribution of results from 2000 repetitions whose standard deviation was indicated by the blue shaded region in the figures.

In order to examine the statistical validity of the machine learning study we produced surrogate data by randomly scrambling the predicted labels of the original data and repeated the procedure of feature selection and classification by cross-validation. The mean of this null population of accuracies is shown by the black curves and its standard deviation by the gray shaded regions in Figs 6 and 7. With a sufficiently large data set and balanced number of targets the null accuracy was expected to approach 50% however the null accuracy in our results was clearly biased above 50%. This was due to the fact that we repeated the *entire study including feature selection* with the surrogate data. Consequently when features were selected according to their correlation with labels, the spuriously correlated features were assigned high priorities, and they generated the higher than 50% null accuracies. As expected the null accuracy declined toward 50% as larger numbers of features were included, as the figure shows. This procedure allowed us to properly identify the part of accuracy that was truly above chance level.

Figs 6 and 7 show that shorter separation channels (1.5 and 2.12 cm) have predictive power distributed over the frontal regions, which may be indicative of correlations between the prediction targets and systemic effects picked up from the layers above the cortex. By contrast the longer separated channels had their peak accuracy occurring with only a few channels (Figs 6C, 6G, 7C, 7D, 7G and 7H), suggesting that the discrimination between levels of task load and skill may be associated with specific PFC areas acting in concert. This finding is consistent with previous results [46] and may be used as a basis for a practical and light-weight brain-computer interface.

In order to further investigate the relative contributions of superficial v cerebral layers we performed SSR, which was designed to minimize the superficial contribution in each longer separated channel [33]. Fig 8 indicates that SSR drastically reduced the accuracy of the longer separated channels. The reduction was greater in the 3 cm separated channels. In the longest separated channels, which probe deeper, post-SSR accuracy was still reasonably high and peaked with a small number of channels (Fig 8B which is to be contrasted with Fig 7D). This confirms that the subjects' level of expertise had a significant effect on the task related PFC activation, as shown by the response from the longest separated channels. The interpretation of the overall reduction in accuracy Fig 8 is less clear, however. This could be due to the fact that the superficial signal was responsible for most of the accuracy (Fig 7C) and the superficial component was mostly eliminated by SSR resulting in Fig 8A. Or, it could be due to the fact

that the cerebral signal had a significant role in the accuracy shown in Fig 7C, but SSR caused much of this to be eliminated because the regressor was taken from channels whose separation (1.5 cm) was not sufficiently short.

We repeated some of the above calculations by including the signals from the motion sensors of the fNIRS headset as additional features. These included three distinct time series from the accelerometer and another three from its gyroscope. Including the motion signals as additional predictors had only minor and unsystematic effects on the accuracy. In addition calculations, we used the motion signals as the only set of features in classification, and this yielded accuracies that were no greater than chance. These results further suggested that motion artifacts had been sufficiently minimized in our preprocessing steps and did not play a role in our results.

Our study had some limitations. The standard optical topography has been used in this study. This method has several disadvantages. First, two dimensional imaging uses sparse arrangements of source and detector optodes and therefore the positions of the measurement channels do not always overlap the real activation foci. Therefore, the spatial resolution of fNIRS imaging is low compared to fMRI. Second, the positions of the measurement channels relative to brain shape vary among subjects, resulting in reduced reliability of comparison among subjects. We plan to use the diffuse optical tomography to solve the resolution problems in the further study.

Short separation channels are essential for accurate fNIRS measurements because they enable the extra-cerebral signal contribution to be regressed from standard separation channels. This reduces the chance that extra-cerebral hemodynamics will be falsely interpreted as functional brain activation. Our results indicates that SSR with 15mm drastically reduced the accuracy This may have been due to the short separation distance is not enough as a regressor. Based on the literature [20, 21]; the optimum short-separation distance is 8.4 mm (vary across the scalp) in the typical adult. The effects of channel separation distance and noncerebral blood flow have been reported in numerous previous studies [20, 47–51].

The number of surgery resident was not enough for statistical analysis. The recruitment was not sufficient due to timeline and schedule problems, logistics issues. We have not investigated the effect of the small group of surgery resident and not compare to students and attending surgeons.

In addition, we have not investigated the longitudional surgical skills learning over time. There is limited research relating to longitudinal surgical skills learning over time. This can be also considered in future studies.

## Conclusion

In this study, we have evaluated prefrontal haemodynamic responses to a set of surgery tasks in two groups: students vs attending surgeons. We quantified the ability of PFC activation to predict the differences in skill and task load between the two groups while focussing on the effects of fNIRS channel separation distance on the results. The high accuracy of results is encouraging and suggest the integration of the strategy developed in this study as a promising approach to design automated, more accurate and objective evaluation methods. Optical imaging is significantly more accurate than current established subjective methods. Our approach brings objectivity, and accuracy in measuring the mental workload and predicting the cognitive load. In particular, this approach may be expanded to robustly identify and predict surgical candidates that may achieve faster learning curves for learning complex surgical skills, and by extension, achieve technical and non-technical mastery with a significantly faster rate than other surgical trainees. In summary, we hope that this neuroimaging approach for objective

quantification for will contribute toward a paradigm change in broad applications, such as surgical certification and assessment, aviation training, and motor skill rehabilitation and therapy.

## Supporting information

**S1 Fig. Accuracy of classification of the NASA-TLX score in task 1.** This figure shows the same results as in Fig 5 but it is for attending subjects. No channels locations are indicated in H since the accuracy remained at chance level.
(TIFF)

**S1 Table. Subject demographics, task completion times and task NASA-TLX scores.**
(DOCX)

**S2 Table. Prefrontal activation associated with high and low NASA-TLX scores for student subjects.** (see Fig 2).
(XLSX)

**S3 Table. Prefrontal activations associated with high and low NASA-TLX scores.** (See Fig 3).
(XLSX)

**S4 Table. Prefrontal activations associated with student and attending subjects for task 1 and task 2.** (See Fig 4).
(XLSX)

## Author Contributions

**Conceptualization:** Hasan Onur Keles.

**Data curation:** Hasan Onur Keles, Canberk Cengiz.

**Formal analysis:** Hasan Onur Keles, Ahmet Omurtag.

**Methodology:** Hasan Onur Keles.

**Software:** Ahmet Omurtag.

**Supervision:** Irem Demiral, Mehmet Mahir Ozmen.

**Validation:** Irem Demiral, Mehmet Mahir Ozmen, Ahmet Omurtag.

**Visualization:** Hasan Onur Keles, Ahmet Omurtag.

**Writing – original draft:** Hasan Onur Keles.

**Writing – review & editing:** Hasan Onur Keles, Ahmet Omurtag.

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
