## [Decision Letter · Decision Letter 0]

3 Dec 2020

PONE-D-20-34607

High Density Optical Neuroimaging predicts surgeons’s subjective experience and skill levels

PLOS ONE

Dear Dr. Keles,

Thank you for submitting your manuscript to PLOS ONE. After careful consideration, we feel that it has merit but does not fully meet PLOS ONE’s publication criteria as it currently stands. Therefore, we invite you to submit a revised version of the manuscript that addresses the points raised during the review process.

Two experts in the field have carefully evaluated the manuscript entitled, "High Density Optical Neuroimaging predicts surgeons’s subjective experience and skill levels". Their comments are appended below.

The first reviewer acknowledged the manuscript is well written leaving some minor methodological concerns regarding machine learning detail.

The second reviewer however raised several major concerns from all the aspects of the manuscript ranging from ‘Data Analysis’, ‘Machine Learning’, ’Results’, and ‘Discussion’. These concerns will be sure to improve and strengthen the manuscript.

I am looking forward receiving your revision according to these critiques and will make judgement.

We look forward to receiving your revised manuscript.

Kind regards,

Manabu Sakakibara, Ph.D.

Academic Editor

PLOS ONE

Journal Requirements:

2. In your Methods section, please provide additional information about the participant recruitment method and the demographic details of your participants.

Please ensure you have provided sufficient details to replicate the analyses such as:

- the recruitment date range (month and year)

- a statement as to whether your sample can be considered representative of a larger population

- a description of how participants were recruited

- descriptions of where participants were recruited and where the research took place.

3. Please list the name and version of any software package used for statistical analysis, alongside any relevant references.

For more information on PLOS ONE's expectations for statistical reporting, please see https://journals.plos.org/plosone/s/submission-guidelines.#loc-statistical-reporting

Reviewers' comments:

Reviewer's Responses to Questions

**Comments to the Author**

1. Is the manuscript technically sound, and do the data support the conclusions?

Reviewer #1: Yes

Reviewer #2: No

2. Has the statistical analysis been performed appropriately and rigorously? 

Reviewer #1: Yes

Reviewer #2: No

3. Have the authors made all data underlying the findings in their manuscript fully available?

Reviewer #1: Yes

Reviewer #2: Yes

4. Is the manuscript presented in an intelligible fashion and written in standard English?

Reviewer #1: Yes

Reviewer #2: No

5. Review Comments to the Author

Reviewer #1: The paper addresses an interesting topic with clinical application. The flow of writing is good and easy to follow. The statistical and machine learning approach are rigorous.

Recommendations: on the machine learning section, it is recommenced to add more details on the feature selection. It is recommended to increase the resolution of the figures.

Reviewer #2: PLOS ONE Review Comments

Summary

In this study, the authors used fNIRS to measure the induced cognitive load on prefrontal cortex of expert and novice surgeons. The authors examined the potential predictive power of these fNIRS measurements for determining the cognitive load as well as the expertise of their participants in performing two laparoscopic surgery tasks: peg transfer and threading.

Major Comments:

Data Analysis:

How did the authors compute the hemoglobin concentrations (e.g., beer-lambert, etc.)? Also, the preprocessing of NIRS time series appear to solely include bandpass filtering without any baseline normalization and detrending (the latter for prevent potential non-stationarity in time series). Another issue is with regards to the use of standard deviation (3 in their case) for artefacts attenuation. This step is quite unconventional and is not (to the best of reviewer’s knowledge) practiced in the literature. The authors are encouraged to consult [1] for a comprehensive review of NIRS preprocessing.

[1] Tak, S. and Ye, J.C., 2014. Statistical analysis of fNIRS data: a comprehensive review. Neuroimage, 85, pp.72-91.

In addition, the authors also appear to use their measured NIRS standard deviation (i.e., after they preprocessed it) for quantification of the brain activation. This is also very unconventional. Specifically, what the authors refer to as “PFC activation” throughout the manuscript is indeed the deviation of the channels’ activation from the average (observed/induced) PFC activity. In fact, such deviation could still be present without any sufficient/significant induced PFC activation by the task. The authors are strongly encouraged to consult [2,3] for measures used for NIRS quantification.

[2] Naseer, N. and Hong, K.S., 2015. fNIRS-based brain-computer interfaces: a review. Frontiers in human neuroscience, 9, p.3.

[3] Keshmiri, S., Sumioka, H., Yamazaki, R. and Ishiguro, H., 2018. Differential entropy preserves variational information of near-infrared spectroscopy time series associated with working memory. Frontiers in neuroinformatics, 12, p.33.

With regard to the use of short-distance channels “to perform superficial signal regression (SSR)” on long-distance channels, it is not clear what methodology/approach the authors adapted in their study. Some of the available approaches are:

[4] Fekete, T., Rubin, D., Carlson, J.M. and Mujica-Parodi, L.R., 2011. The NIRS analysis package: noise reduction and statistical inference. PloS one, 6(9), p.e24322.

[5] Zhang, Y., Brooks, D.H., Franceschini, M.A. and Boas, D.A., 2005. Eigenvector-based spatial filtering for reduction of physiological interference in diffuse optical imaging. Journal of biomedical optics, 10(1), p.011014.

[6] Kohno, S., Miyai, I., Seiyama, A., Oda, I., Ishikawa, A., Tsuneishi, S., Amita, T. and Shimizu, K., 2007. Removal of the skin blood flow artifact in functional near-infrared spectroscopic imaging data through independent component analysis. Journal of biomedical optics, 12(6), p.062111.

[7] Haeussinger, F.B., Dresler, T., Heinzel, S., Schecklmann, M., Fallgatter, A.J. and Ehlis, A.C., 2014. Reconstructing functional near-infrared spectroscopy (fNIRS) signals impaired by extra-cranial confounds: an easy-to-use filter method. NeuroImage, 95, pp.69-79.

[8] Gagnon, L., Perdue, K., Greve, D.N., Goldenholz, D., Kaskhedikar, G. and Boas, D.A., 2011. Improved recovery of the hemodynamic response in diffuse optical imaging using short optode separations and state-space modeling. Neuroimage, 56(3), pp.1362-1371.

[9] Keshmiri, S., Sumioka, H., Okubo, M. and Ishiguro, H., 2019. An Information-Theoretic Approach to Quantitative Analysis of the Correspondence Between Skin Blood Flow and Functional Near-Infrared Spectroscopy Measurement in Prefrontal Cortex Activity. Frontiers in neuroscience, 13, p.79.

Analysis Steps with Machine Learning:

Although the authors mentioned the use of signal’s standard deviation for quantifying the brain activation, they then switched to its mean for ML-based classification (page 8: “For this purpose feature matrices were built with each row (an observation) representing the mean of the prefrontal activation over an episode”). Such inconsistencies and mixing of measures/metrics make quite difficult to realize the potential underlying property of the signal based on which the results have been derived.

The authors also mentioned that (page 9) “The feature types were prioritised by using the Pearson correlation between the observations and the labels, a standard feature-selection technique” – Do authors refer to the channels as features? If so, this step actually decided on which subset of channels to be used as inputs to their ML model.

The authors continued by explaining that (page 9) “We then chose a small group of features from the prioritised list and used it to train Support Vector Machines (SVM) with linear kernels.” – How did the authors decide this “small group of features?” Did they apply such utilities as “features importance” that are available through ML libraries? If so, what criterion/criteria was/were used to determine the level of significance of feature scores? (while using the term “feature,” I am assuming “selected channels” as per authors’ earlier explanation).

The authors also used 5-fold cross-validation for testing the accuracy of their model. As the authors explained, every participant in their study participated in two different tasks (i.e., Peg Transfer and

Threading). As such, did the authors ensure that data from the same individual were not present in both train and test sets while applying 5-fold cross-validation? This is an important issue while performing such analyses since data from the same individuals should not be expected to be highly different between the two tasks which could, in turn, results in overestimation of the model’s accuracy.

Result:

As one of their hypotheses, the authors stated (page 9) “that there would be differences in the activations due to the different sampling depths of the channels with different separation distances.” – The effect of channel separation and skin- other than cortical-blood-flow is a well-studied subject. In fact, short-distance channels are not expected to represent cortical activity. Similarly, channels with distances larger than 3.5 cm have been also generally accepted to not produce reliable results due to the absorption of optical signals as it penetrates deeper to cortical tissues. Please consult [5-9] above for more in-depth results and discussion on this matter.

The authors’ statement (page 10) “Figure 2 shows that in the Student subjects who had experienced higher task load also had higher PFC activations.” that is quite repeated throughout the manuscript is not really valid since the authors used the standard deviation of the signal. In other words, these quantities are how individuals’ PFC activity deviated from the averaged observed/induced PFC activation in their study.

With regard to results’ presentation, the authors sufficed to such statements as (e.g., page 10) “The difference between high and low load subjects were statistically significant in both Tasks in the case of the shortest (1.5 cm, A and E) and the normal separation (3 cm, C and G) channels.” while referring to the figures 2 through 7 without providing any descriptive statistics for their results. Precisely, it is not clear how the authors determined these results “were statistically significant?” What type of tests did they apply? What were the p-values, test-statistics, mean, standard deviation, confidence intervals, and effect-sizes associated with these tests? Did the authors corrected their p-values while determining their significance (e.g., Bonferroni, FDR, etc.)?

The authors stated that (page 11) “In addition, for the student subjects there was a pronounced asymmetry in the case of the deepest sampling channel (D and H), the activation on the left being significantly higher than on the right.” – The reviewer encourages the authors to consult the studies related to the effect of short/long-distance channels on NIRS measurements that are listed above. In particular, the “deepest sampling channel” in the present study could fall within the range that is considered not suitable for studying the cortical activation.

With regard to the source localization presented in Figure 5, the authors used such statements as (page 11) “there is a hint of high activation localized near the top left …” – Such assertion are not justified unless the authors provide statistical evidence for the possibility of such activations that differ from the other regions.

Above shortcomings with regard to the results’ representation also apply to the case of the results pertinent to ML-based results.

Discussion:

The authors stated that (page 13) “This difference, visible in most channel separations, was statistically significant in the 1.5 cm and most of the 3 cm separated channels (Figure 2).” – The authors did not present sufficient statistics for this claim (only presenting figures).

They also stated that (page 13) “in skilled subjects in the correlation of PFC activation with subjective task” however, the reviewer could not find/see these correlation analyses.

Another issue is with regard to hemispheric differences that authors referred to (page 14) “We found that response was greater in the left PFC of students (Figure 4D and H), ...” – Unless the authors perform statistical tests, such claims are not truly founded.

Other Comments:

The language of the manuscript requires a thorough auditing and proofread as it is not easy to follow and comprehend the study.

The quality of figures are very low and must be improved.

Please also break your Results Section into different subsections (e.g., one for test of significant differences, another for ML-based results, etc.) to help reader better follow and understand the results.

6. PLOS authors have the option to publish the peer review history of their article (what does this mean?). If published, this will include your full peer review and any attached files.

Reviewer #1: No

Reviewer #2: **Yes: **Soheil Keshmiri

---

## [Author Response · Author response to Decision Letter 0]

18 Dec 2020

Reviewer #1: The paper addresses an interesting topic with clinical

application. The flow of writing is good and easy to follow. The

statistical and machine learning approach are rigorous.

Recommendations: on the machine learning section, it is recommenced to

add more details on the feature selection. It is recommended to increase

the resolution of the figures.

OUR RESPONSE: We have new details on feature selection in the new version. The new figures have high resolution.

Reviewer #2: PLOS ONE Review Comments

Summary

In this study, the authors used fNIRS to measure the induced cognitive

load on prefrontal cortex of expert and novice surgeons. The authors

examined the potential predictive power of these fNIRS measurements for

determining the cognitive load as well as the expertise of their

participants in performing two laparoscopic surgery tasks: peg transfer

and threading.

Major Comments:

Data Analysis:

-- How did the authors compute the hemoglobin concentrations (e.g.,

beer-lambert, etc.)? 

OUR RESPONSE: The modified Beer-Lambert Law (MBLL) was applied to compute the concentration changes of HbO and HbR. The differential path length factor (DPF) values of 780 nm and 850nm are 5.075 and 4.64 in respectively. We provide an explanation of how we compute the haemoglobin concentrations in the updated preprocessing section. We have also cited the paper by Delpy 1988, Choi 2013 and Boas 2000. 

-- Also, the preprocessing of NIRS time series appear

to solely include bandpass filtering without any baseline normalization

and detrending (the latter for prevent potential non-stationarity in

time series). 

OUR RESPONSE: In our case the highpass filter (0.01 Hz) eliminated all baseline drift and any slow features (longer than 100 s). This was sufficient for the purposes of this study as our features were standard deviations computed within 10 s windows. The 10 second standard deviation was a mean subtracted measure hence not affected by any remaining slow components in the signal to our knowledge. The preprocessing stages and their justification are explained in greater detail in the new, expanded Data Analysis section.

-- Another issue is with regards to the use of standard

deviation (3 in their case) for artefacts attenuation. This step is

quite unconventional and is not (to the best of reviewer's knowledge)

practiced in the literature. The authors are encouraged to consult [1]

for a comprehensive review of NIRS preprocessing.

[1] Tak, S. and Ye, J.C., 2014. Statistical analysis of fNIRS data: a

comprehensive review. Neuroimage, 85, pp.72-91.

OUR RESPONSE: The standard deviation has in fact been used for artefact attenuation, for example Scholkmann 2010. In the next version of the manuscript we provide extensive description and justification of our method (together with a new figure). We have also cited the paper by Tak and Ye 2014.

-- In addition, the authors also appear to use their measured NIRS standard

deviation (i.e., after they preprocessed it) for quantification of the

brain activation. This is also very unconventional. Specifically, what

the authors refer to as "PFC activation" throughout the manuscript is

indeed the deviation of the channels' activation from the average

(observed/induced) PFC activity. In fact, such deviation could still be

present without any sufficient/significant induced PFC activation by the

task. The authors are strongly encouraged to consult [2,3] for measures

used for NIRS quantification.

[2] Naseer, N. and Hong, K.S., 2015. fNIRS-based brain-computer

interfaces: a review. Frontiers in human neuroscience, 9, p.3.

[3] Keshmiri, S., Sumioka, H., Yamazaki, R. and Ishiguro, H., 2018.

Differential entropy preserves variational information of near-infrared

spectroscopy time series associated with working memory. Frontiers in

neuroinformatics, 12, p.33.

OUR RESPONSE: The variance (closely related to the standard deviation used by our study) was described as one of the features available for use in fNIRS analysis by Naseer and Hong 2015. We have cited this paper in the new version of the manuscript. 

Evoked hemodynamic response in a 10 s window this may be in the form of a rise in the oxyhemoglobin concentration followed by a dip. The within-window standard deviation is a sensitive measure for capturing this response. We provide an explanation of how in our study the standard deviation can be legitimately considered as a measure of PFC activation in the new subsection Feature extraction and selection

-- With regard to the use of short-distance channels "to perform

superficial signal regression (SSR)" on long-distance channels, it is

not clear what methodology/approach the authors adapted in their study.

Some of the available approaches are:

[4] Fekete, T., Rubin, D., Carlson, J.M. and Mujica-Parodi, L.R., 2011.

The NIRS analysis package: noise reduction and statistical inference.

PloS one, 6(9), p.e24322.

[5] Zhang, Y., Brooks, D.H., Franceschini, M.A. and Boas, D.A., 2005.

Eigenvector-based spatial filtering for reduction of physiological

interference in diffuse optical imaging. Journal of biomedical optics,

10(1), p.011014.

[6] Kohno, S., Miyai, I., Seiyama, A., Oda, I., Ishikawa, A., Tsuneishi,

S., Amita, T. and Shimizu, K., 2007. Removal of the skin blood flow

artifact in functional near-infrared spectroscopic imaging data through

independent component analysis. Journal of biomedical optics, 12(6),

p.062111.

[7] Haeussinger, F.B., Dresler, T., Heinzel, S., Schecklmann, M.,

Fallgatter, A.J. and Ehlis, A.C., 2014. Reconstructing functional

near-infrared spectroscopy (fNIRS) signals impaired by extra-cranial

confounds: an easy-to-use filter method. NeuroImage, 95, pp.69-79.

[8] Gagnon, L., Perdue, K., Greve, D.N., Goldenholz, D., Kaskhedikar, G.

and Boas, D.A., 2011. Improved recovery of the hemodynamic response in

diffuse optical imaging using short optode separations and state-space

modeling. Neuroimage, 56(3), pp.1362-1371.

[9] Keshmiri, S., Sumioka, H., Okubo, M. and Ishiguro, H., 2019. An

Information-Theoretic Approach to Quantitative Analysis of the

Correspondence Between Skin Blood Flow and Functional Near-Infrared

Spectroscopy Measurement in Prefrontal Cortex Activity. Frontiers in

neuroscience, 13, p.79.

OUR RESPONSE: In fact we used the method in Gagnon et al 2011. However due to an error in the manuscript the reference number showed another paper. This has now been fixed and we cite the Gagnon et al paper in the new subsection Feature extraction and selection.

Analysis Steps with Machine Learning:

-- Although the authors mentioned the use of signal's standard deviation

for quantifying the brain activation, they then switched to its mean for

ML-based classification (page 8: "For this purpose feature matrices were

built with each row (an observation) representing the mean of the

prefrontal activation over an episode"). Such inconsistencies and mixing

of measures/metrics make quite difficult to realize the potential

underlying property of the signal based on which the results have been

derived.

OUR RESPONSE: This in fact a misunderstanding due an unclear statement in the manuscript. The features we used by the standard deviations that were averaged over an experimental episode. Thus we did not switch to a new metric, but only averaged our metric over an episode. (The reason for doing so is related to another probing question raised by the reviewer, namely whether we included data from the same subject in both training and test sets. By averaging over an episode we insured that there was only one row per subject in the feature matrix.) We have changed the unclear statement in the subsection Classification by inserting the phrase “representing the standard deviations averaged over an episode”.

-- The authors also mentioned that (page 9) "The feature types were

prioritised by using the Pearson correlation between the observations

and the labels, a standard feature-selection technique" - Do authors

refer to the channels as features? If so, this step actually decided on

which subset of channels to be used as inputs to their ML model.

OUR RESPONSE: Yes, each channel contributes a separate column in the feature matrix.

-- The authors continued by explaining that (page 9) "We then chose a small

group of features from the prioritised list and used it to train Support

Vector Machines (SVM) with linear kernels." - How did the authors decide

this "small group of features?" Did they apply such utilities as

"features importance" that are available through ML libraries? If so,

what criterion/criteria was/were used to determine the level of

significance of feature scores? (while using the term "feature," I am

assuming "selected channels" as per authors' earlier explanation).

OUR RESPONSE: The Pearson correlation between a feature and the labels is a way of estimating the extent to which that feature will be helpful in classifying the labels. We have used this technique in previous studies e.g. Omurtag et al 2017 which provides references for it. But in the new version of the manuscript we provided additional details and referenced two other methods Minimum Redundance and Maximum Relevance and the Chi-square tests, available through Matlab. We explored these methods but they are more involved than the Pearson correlation without any clear benefits, hence were not adopted in our study.

-- The authors also used 5-fold cross-validation for testing the accuracy

of their model. As the authors explained, every participant in their

study participated in two different tasks (i.e., Peg Transfer and

Threading). As such, did the authors ensure that data from the same

individual were not present in both train and test sets while applying

5-fold cross-validation? This is an important issue while performing

such analyses since data from the same individuals should not be

expected to be highly different between the two tasks which could, in

turn, results in overestimation of the model's accuracy.

OUR RESPONSE: Thanks for pointing out this important issue. We ensured that the feature matrix contained only one row per subject for each classification problem. Therefore the training and test sets did not share data from the same subject. This is clarified in the new subsection Classification.

Result:

-- As one of their hypotheses, the authors stated (page 9) "that there

would be differences in the activations due to the different sampling

depths of the channels with different separation distances." - The

effect of channel separation and skin- other than cortical-blood-flow is

a well-studied subject. In fact, short-distance channels are not

expected to represent cortical activity. Similarly, channels with

distances larger than 3.5 cm have been also generally accepted to not

produce reliable results due to the absorption of optical signals as it

penetrates deeper to cortical tissues. Please consult [5-9] above for

more in-depth results and discussion on this matter.

OUR RESPONSE: We have added new statement and new references in the 3rd paragraph of the Discussion section reminding the reader of degraded signal-to-noise associated with longer source-detector separations. The long separation we have used is 3.35 cm, which is well within the range of 25-35 mm which is considered typical in several fNIRS studies, which we provide references for. Although we realize that, as indicated by the reviewer, one should try to avoid the longer end of this range in order to avoid noise, as we have done.

-- The authors' statement (page 10) "Figure 2 shows that in the Student

subjects who had experienced higher task load also had higher PFC

activations." that is quite repeated throughout the manuscript is not

really valid since the authors used the standard deviation of the

signal. In other words, these quantities are how individuals' PFC

activity deviated from the averaged observed/induced PFC activation in

their study.

OUR RESPONSE: The reviewer correctly points out that the observed PFC activation is typically based on haemoglobin concentration changes. However we believe that for the purposes of this study, their standard deviation is an equally valid meaning of the term activation, because (with proper preprocessing) greater evoked hemodynamic responses resulted in greater standard deviations. In this version of the manuscript we have clarified this definition of “activation” in the beginning of the new subsection “Feature extraction and selection” in Methods with the sentence: “Because greater evoked hemodynamic response tends to increase the standard deviation of the signal in a window, we took these values as the measure of the prefrontal activation.” Because of this close relationship, introducing a different term (other than activation) might have been confusing to the readers. We respectfully would like to keep this term, as it has now been clearly defined in the new version of the manuscript; it does not dramatically differ from the typical interpretation, and it provides a concise and coherent way of describing our results.

-- With regard to results' presentation, the authors sufficed to such

statements as (e.g., page 10) "The difference between high and low load

subjects were statistically significant in both Tasks in the case of the

shortest (1.5 cm, A and E) and the normal separation (3 cm, C and G)

channels." while referring to the figures 2 through 7 without providing

any descriptive statistics for their results. Precisely, it is not clear

how the authors determined these results "were statistically

significant?" What type of tests did they apply? What were the p-values,

test-statistics, mean, standard deviation, confidence intervals, and

effect-sizes associated with these tests? Did the authors corrected

their p-values while determining their significance (e.g., Bonferroni,

FDR, etc.)?

OUR RESPONSE: Thanks for pointing out this gap in our manuscript. We have added new material under the new subsection Statistical analysis under the Methods section. We describe at length our approach there that let us determine the statistical significance of our results. In the new version we have also introduced Bonferroni correction to adjust the significance cut-off for the p-values. This new cut-off slightly affected the figures, hence we have introduced new versions of the box plot figures, where a few of the previously significant results appear without an asterisk. However the majority of the significance results still remain. 

--The authors stated that (page 11) "In addition, for the student subjects

there was a pronounced asymmetry in the case of the deepest sampling

channel (D and H), the activation on the left being significantly higher

than on the right." - The reviewer encourages the authors to consult the

studies related to the effect of short/long-distance channels on NIRS

measurements that are listed above. In particular, the "deepest sampling

channel" in the present study could fall within the range that is

considered not suitable for studying the cortical activation.

OUR RESPONSE: In the 3rd of paragraph of the Discussion section we address this issue and provide new references.

-- With regard to the source localization presented in Figure 5, the

authors used such statements as (page 11) "there is a hint of high

activation localized near the top left …" - Such assertion are not

justified unless the authors provide statistical evidence for the

possibility of such activations that differ from the other regions.

OUR RESPONSE: We thank the reviewer for pointing out this oversight in the manuscript. We have inserted in the new version a new paragraph immediately after Figure 5 which explains how to assess the statistical significance of Figure 5 based on the corresponding subplots of Figure 4. The greater activation in student subjects in the left PFC is significant by this analysis, as explained in the new paragraph.

Above shortcomings with regard to the results' representation also apply

to the case of the results pertinent to ML-based results.

Discussion:

-- The authors stated that (page 13) "This difference, visible in most

channel separations, was statistically significant in the 1.5 cm and

most of the 3 cm separated channels (Figure 2)." - The authors did not

present sufficient statistics for this claim (only presenting figures).

OUR RESPONSE: This gap in the manuscript has now been addressed by fully describing the statistical analysis in the new subsection Statistical analysis under the section Method, and by the new version of Figure 2 which includes clear indications of which group differences are significantly different.

-- They also stated that (page 13) "in skilled subjects in the correlation

of PFC activation with subjective task" however, the reviewer could not

find/see these correlation analyses.

OUR RESPONSE: To improve clarity this sentence has now been modified so that it reads: “This change in skilled subjects’ PFC response relative to that of unskilled ones may have been due to…”. 

-- Another issue is with regard to hemispheric differences that authors

referred to (page 14) "We found that response was greater in the left

PFC of students (Figure 4D and H), ..." - Unless the authors perform

statistical tests, such claims are not truly founded.

OUR RESPONSE: We believe this valid critique has now been answered in the new version of the manuscript since we have a new section about the statistical analysis and we show the significant differences clearly on the figures. The difference mentioned in this comment is indeed statistically significant according to our analysis.

Other Comments:

-- The language of the manuscript requires a thorough auditing and

proofread as it is not easy to follow and comprehend the study.

OUR RESPONSE: We have introduced numerous clarifications and some stylistic edits (in addition to the corrections listed above).

-- The quality of figures are very low and must be improved.

OUR RESPONSE: The quality of figures are improved. We used PACE diagnostic tool to ensure that figures meet PLOS One requirements. 

.

-- Please also break your Results Section into different subsections (e.g.,

one for test of significant differences, another for ML-based results,

etc.) to help reader better follow and understand the results.

OUR RESPONSE: We have introduced the new subsection titles “Group differences” and “Machine learning results” in the Results section.

---

## [Decision Letter · Decision Letter 1]

19 Jan 2021

PONE-D-20-34607R1

High Density Optical Neuroimaging predicts surgeons’s subjective experience and skill levels

PLOS ONE

Dear Dr. Keles,

Thank you for submitting your manuscript to PLOS ONE. After careful consideration, we feel that it has merit but does not fully meet PLOS ONE’s publication criteria as it currently stands. Therefore, we invite you to submit a revised version of the manuscript that addresses the points raised during the review process.

The two original reviewers have carefully reviewed the revised manuscript. Their comments are appended below. Both of them acknowledged the improvement of the manuscript still leaving several concerns which should be considered before publication.

I will make the decision after receipt of replies to each critique and the necessary revision.

We look forward to receiving your revised manuscript.

Kind regards,

Manabu Sakakibara, Ph.D.

Academic Editor

PLOS ONE

Reviewers' comments:

Reviewer's Responses to Questions

**Comments to the Author**

1. If the authors have adequately addressed your comments raised in a previous round of review and you feel that this manuscript is now acceptable for publication, you may indicate that here to bypass the “Comments to the Author” section, enter your conflict of interest statement in the “Confidential to Editor” section, and submit your "Accept" recommendation.

Reviewer #1: (No Response)

Reviewer #2: All comments have been addressed

2. Is the manuscript technically sound, and do the data support the conclusions?

Reviewer #1: Yes

Reviewer #2: Yes

3. Has the statistical analysis been performed appropriately and rigorously? 

Reviewer #1: Yes

Reviewer #2: No

4. Have the authors made all data underlying the findings in their manuscript fully available?

Reviewer #1: Yes

Reviewer #2: Yes

5. Is the manuscript presented in an intelligible fashion and written in standard English?

Reviewer #1: Yes

Reviewer #2: Yes

6. Review Comments to the Author

Reviewer #1: The paper is on an interesting topic, well written and method is rigorous. It would be better to add other ML metrics in the analysis to have a better comparison as well.

Needs to be fixed: Image resolution is needed to be higher. Figures are not captioned.

Reviewer #2: Thank you for taking your time and addressing my comments. The quality of the manuscript is substantially improved. However, there is still one point that the reviewer would like authors to address.

Although the authors now present the statistical test that they have used (i.e., non-parametric Wilcoxon signed-rank test), they still miss presenting the results of these tests. The reviewer realizes that the authors provided figures for their tests. However, these figures on their own are not sufficient. Please provide the numerical results of the test statistics associated with these figures (e.g., in tabular formats) providing test-statistics, p-values, M/SD, and confidence interval of your tests.

7. PLOS authors have the option to publish the peer review history of their article (what does this mean?). If published, this will include your full peer review and any attached files.

Reviewer #1: No

Reviewer #2: **Yes: **Soheil Keshmiri

---

## [Author Response · Author response to Decision Letter 1]

23 Jan 2021

Dear Prof. Manabu Sakakibara, 

Thank you for sending the reviewer comments for our revised manuscript. We are very happy to see that they find the revised version greatly improved.

Regarding Reviewer 1:

We used PACE diagnostic tool to ensure that figures meet PLOS One requirements. If there is a remaining problem with the figures, we will be happy to address them promptly. In the revised submission, figure captions were inserted in the text of manuscript. Because of PLOS One requirements, we did not include captions as part of the figure files. 

Regarding Reviewer 2:

We appreciate this reviewer’s close scrutiny of our revised MS. The only remaining criticism from this reviewer is that the values in the results section should be given (in addition to the figures). We have now given the values as tables in Supp. Information section.

We believe the new version fully meets all reviewers’ requests; however we will be happy to address any remaining issues with the manuscript.

Sincerely,

Hasan Onur Keles, PhD

---

## [Decision Letter · Decision Letter 2]

2 Feb 2021

High Density Optical Neuroimaging predicts surgeons’s subjective experience and skill levels

PONE-D-20-34607R2

Dear Dr. Keles,

We’re pleased to inform you that your manuscript has been judged scientifically suitable for publication and will be formally accepted for publication once it meets all outstanding technical requirements.

Kind regards,

Manabu Sakakibara, Ph.D.

Academic Editor

PLOS ONE

Additional Editor Comments (optional):

Since the previous reviewer #1 is not available this time, I, this Academic Editor, and the reviewer #2 have carefully evaluated the revision #2. The reviewer #2 and I am totally satisfied with the revised manuscript. The revision has satisfactorily improved, thus the manuscript is accepted for publication in PLOS ONE.

Reviewers' comments:

Reviewer's Responses to Questions

**Comments to the Author**

1. If the authors have adequately addressed your comments raised in a previous round of review and you feel that this manuscript is now acceptable for publication, you may indicate that here to bypass the “Comments to the Author” section, enter your conflict of interest statement in the “Confidential to Editor” section, and submit your "Accept" recommendation.

Reviewer #2: All comments have been addressed

2. Is the manuscript technically sound, and do the data support the conclusions?

Reviewer #2: Yes

3. Has the statistical analysis been performed appropriately and rigorously? 

Reviewer #2: Yes

4. Have the authors made all data underlying the findings in their manuscript fully available?

Reviewer #2: Yes

5. Is the manuscript presented in an intelligible fashion and written in standard English?

Reviewer #2: Yes

6. Review Comments to the Author

Reviewer #2: (No Response)

7. PLOS authors have the option to publish the peer review history of their article (what does this mean?). If published, this will include your full peer review and any attached files.

Reviewer #2: **Yes: **Soheil Keshmiri

---

## [Editor Report · Acceptance letter]

5 Feb 2021

PONE-D-20-34607R2 

High Density Optical Neuroimaging predicts surgeons’s subjective experience and skill levels. 

Dear Dr. Keles:

I'm pleased to inform you that your manuscript has been deemed suitable for publication in PLOS ONE. Congratulations! Your manuscript is now with our production department. 

Kind regards, 

on behalf of

Dr. Manabu Sakakibara 

Academic Editor

PLOS ONE